# Factors predicting participation in organised sports during adolescence: A two-year longitudinal perspective

**Gwennyth E. Spruijtenburg** *, Femke van Abswoude, Hidde Bekhuis, Bert Steenbergen

Behavioural Science Institute (BSI), Radboud University, Nijmegen, Netherlands

* gwennyth.spruijtenburg@ru.nl

**Data Availability Statement:** The minimal dataset is available from Open Science Framework at https://doi.org/10.17605/OSF.IO/7AD4Y. All other data files are available from DANS EASY at https://doi.org/10.17026/SS/LA3KIZ.

## Abstract

### Background

Participation in youth sports is a major determinant of current and future health, yet participation rates are declining. It is of great importance to examine the factors that potentially influence adolescents' levels of participation in organised sports.

### Aim

First, we examined change in participation over two years from the start of secondary education. Second, we examined whether a combination of factors (i.e. motives, barriers, perceived competence, encouragement and motor skills) based on the Youth Physical Activity Promotion model was associated with participation after one year. Third, we examined whether the associations between each of the factors and participation in organised sports changed over time.

### Methods

Data were collected on three occasions between October 2020 and November 2022 using questionnaires, motor skill test items and anthropometric measurements. We analysed data from 204 Dutch adolescents (11 to 14 years) using dependent $t$ tests, Cochran's $Q$ test and multilevel linear fixed effects regression models.

### Results

First, participation in organised sports was stable during the first years of secondary education. Second, adolescents who reported higher motives, higher perceptions of competence and more encouragement spent more time in organised sports one year later. Barriers and motor skills did not predict participation. Third, relationships of each of the six factors with participation remained stable over time.

### Conclusion

Although participation in organised sports is often considered to decline during the first years of secondary education, it can be stable for a significant portion of the population.

**Funding:** This research was supported by a grant (TRIAL) from the Dutch Research Council (NWO), grant number NWA.1160.18.249. The funders had no role in study design, data collection and analysis, decision to publish, or preparation of the manuscript.

**Competing interests:** The authors have declared that no competing interests exist.

Motives, perceived competence, and encouragement seem crucial to stimulate adolescents' participation in organised sports.

## Introduction

Participation in organised sports during adolescence has been shown to positively influence physical, psychological and social health [1]. Research suggests that children and adolescents who participate in sports have better psychological and social health, above and beyond those who participate in other forms of leisure-time physical activity [1]. Moreover, the benefits of youth sports are long-lasting as evidenced by a study showing that children and adolescents who participate in sports more regularly and for at least 3 years were more likely to be physically active in adulthood [2]. Participation in youth sports is thus a major determinant of current and future health. Yet, over a third of school-aged children do not participate in organised sports and the participation rates of this group decline with age [1, 3–6]. To counter this trend, there is a need for tailor-made interventions and programs that stimulate young people to participate in organised sports regularly and to maintain their level of participation in sports over time. This in turn contributes to a long-term active lifestyle. It is therefore of great importance to examine the factors that influence adolescents' levels of participation in organised sports.

Much of the research to date that examined participation in sports has been cross-sectional or retrospective in nature (see [7–9]). Although they have led to important insights into the factors that influence participation in youth sports, cross-sectional and retrospective studies are limited in their ability to determine the causal relationships between influencing factors and participation in sports over time. In order to address this causality aspect, prospective longitudinal studies are needed. In the present study, we will therefore follow a group of 204 Dutch adolescents over time during their first years of secondary education. We will analyse data from the same group on three occasions that are 12 months apart, i.e. at the start of adolescents' first, second and third year of secondary education. Both adolescents' participation in organised sports as well as potential influencing factors will be assessed.

Our study is guided by the Youth Physical Activity Promotion (YPAP) model [10]. This socioecological model categorizes the factors that may affect participation in youth sports into three constructs: predisposing, reinforcing and enabling. The predisposing construct represents adolescents' perceptions and motivation towards sports and consists of two sub-constructs: "Am I able?" (e.g. perceptions of motor competence) and "Is it worth it?" (e.g. desirable outcomes of participation). Furthermore, the reinforcing construct represents social influences (e.g. parent, friend, teacher, and coach) and the enabling construct represents environmental variables (e.g. access to sports equipment) and other variables that allow adolescents to be physically active (e.g. motor skills). According to the YPAP model, all three constructs may directly encourage or discourage adolescents' participation, whereas the reinforcing and enabling constructs may also influence participation indirectly through the predisposing construct. The YPAP model hereby offers a great opportunity to study not only the individual role but also the combined role of individual and environmental factors in relation to adolescents' participation in organised sports.

The findings of several systematic reviews indicate that specific factors at the predisposing, reinforcing and enabling levels of the YPAP model stand out as particularly important for continued participation in and dropout from youth sports [7–9, 11]. At the predisposing level,

potential outcomes of sport such as enjoyment or a lack thereof, and perceptions of competence are considered key factors that influence persistent participation [8, 9] and dropout [7]. At the reinforcing level, support from parents, coaches and peers has been shown to be one of the most important influencing factors of continued participation in youth sports [8, 9, 11]. At the enabling level, reviews showed that sports skills were consistently related to the risk of dropout [11] and continued participation [9]. Taken collectively, these systematic reviews provide support for the predisposing, reinforcing and enabling constructs of the YPAP model. Nevertheless, because their findings were for the most part based on studies with cross-sectional or retrospective designs that focused only on one sport, an important next step is to investigate the validity of the constructs of the YPAP model prospectively over time and across a range of sports.

Only a few studies prospectively examined potential influencing factors in relation to participation in sports across different types of sports [12–14]. An important outcome of these studies is that combinations of factors (i.e. gender, enjoyment, peer acceptance, parental support) are predictive of participation in sports over time. However, none of these studies included factors at all levels of the YPAP model. In addition, motor skills were included as a potential influencing factor in only one of these prospective studies, whereas previous reviews have shown that this is an important aspect of continued participation in certain sports [9, 11]. It is therefore key to simultaneously investigate the impact of factors at all levels of the YPAP model in order to account for the multidimensional influences that interact to determine participation in organised sports.

In this paper, we firstly examine change in participation in organised sports over two years from the start of secondary education. In line with previous large-scale studies, we expect that adolescents' levels of participation will decline [1, 3–6]. Secondly, we examine whether a combination of factors based on the YPAP model (i.e. motives, barriers, perceived competence, encouragement, and motor skills) was associated with time spent in organised sports one year later during adolescence. We hypothesize that motives, perceived competence, encouragement and motor skills will be positively related to participation in organised sports one year later, whereas barriers will be negatively related to participation one year later [7–9, 11]. Finally, we examine whether the relationships between the factors and participation in organised sports change over time. We hereby build on the insights of our previous cross-sectional study in which we found motives, perceived competence, encouragement and motor skills were associated with a higher likelihood of participating in organised sports in first-year secondary school students [15].

## Methods

### Study design

This longitudinal study is part of a larger research project (Transitions Into Active Living, TRIAL) that focuses on changes in physical activity behaviour during key life transitions. Secondary school students were enrolled in the study in October 2020, shortly after they transitioned from primary to secondary education, and were followed until November 2022, shortly after they transitioned to their third year of secondary education. Three waves of data collection took place: Wave 1 (October/November 2020), Wave 2 (October/November 2021) and Wave 3 (October/November 2022), when adolescents were at the start of their first, second and third year of secondary education, respectively. These data waves were the only ones that included every measure of the current study. This study was approved by the Ethics Committee of the Faculty of Social Sciences at Radboud University (ECSW-2020-107).

## Study sample

We selected a convenience sample of eight secondary schools with which our research team established a working relationship. These schools were located in two provinces in the eastern part of the Netherlands. We informed the physical education teachers via email and invited them to participate with their school in June 2020. Five schools agreed to participate. We subsequently sent information letters and informed consent forms on paper and via email to all first-year students of these schools (n = 1127) and their parents. The recruitment period for this study was from September 9, 2020, to the end of October of the same year. Out of the potential participants, a total of 530 adolescents (50.9% boys) had parental written consent to participate in the study and were requested to give written consent themselves.

Adolescents were included only in the present analyses if they had complete data on the independent variables (i.e. motives, barriers, perceived competence, encouragement and motor skills) on Wave 1 and 2 and on the dependent variable (i.e. hours of participation in organised sports) on Wave 2 and Wave 3. This resulted in a final analysis sample of 204 adolescents (51.5% boys) aged 11 to 14 years ($M = 12.5$, $SD = 0.4$) with a mean body mass index (BMI) of 18.2 kg/m$^2$ ($SD = 2.4$) at the start of the study. At Wave 1, complete data were available for 390 students. By Wave 2, this number had decreased to 285 (73.1% of the original 390). By Wave 3, the number of students with complete data further decreased to 204 (52.3% of the original 390). Therefore, the dropout rate from the initial sample was 26.9% by Wave 2 and 47.7% by Wave 3. Common reasons for dropout included repeating a grade, moving to a different location, no longer wanting to participate, and being ill on the day of testing. The adolescents included in the analysis were enrolled in different educational paths at the start of the study: 9% in pre-vocational, 12% in pre-vocational/higher general, 24% in pre-vocational/higher general/pre-university, 34% in higher general/pre-university, and 21% in pre-university. Eighty-nine percent of the included adolescents were native Dutch and 11% was non-native Dutch. There were no differences by gender, age, BMI and ethnicity between adolescents included and excluded from the study.

## Assessments

**Background measures.** Adolescents' age, gender, ethnicity and BMI were assessed at the start of the study. Students self-reported their age and gender. For ethnicity, we followed the definition of Statistics Netherlands [16, 17]. As such, adolescents were asked to report their parents' countries of birth. Based on this information, they were categorized as native Dutch when both their parents were born in the Netherlands [16] or as non-native Dutch when at least one of their parents was born outside the Netherlands [17]. Trained research assistants measured adolescents' height (Seca stadiometer) and weight (Seca scale). Based on height and weight, we calculated BMI (kg/m$^2$).

**Organised sports.** Organised sports were defined as any sport that can be practiced in sport clubs (e.g. soccer and tennis) and fitness centres and any other sport that is led by a trainer or coach and includes formal practice (e.g. boot camp). Participation in organised sports (yes/no) was measured using four items that were based on the validated Flemish Physical Activity Questionnaire (FPAQ) [18]. The first item asked adolescents whether they participated in sports. If so, a second item assessed the sports in which they participated. Students could fill out a maximum of three sports. For each of these sports, two items assessed whether the sport was organised, e.g. whether students practiced the sport in a sport club or fitness centre and whether they practiced the sport under the supervision of a trainer or coach. The sport was classified as organised when at least one of both items was answered with "yes". Participation in organised sports was classified as "yes" when at least one of the sports was organised.

Hours of participation in organised sports per week was measured using a maximum of three additional items, one for each organised sport that the students had filled out. Based on these items, we calculated the total hours of participation in organised sports per week. If a student did not participate in any organised sport, then the hours of participation in organised sports was set at zero. Two outliers with regard to hours of participation (i.e. 90 and 230 hours) were removed from the dataset as these values are practically impossible (highest remaining value was 24 hours).

**Motives.**   Motives referred to the reasons why students participate or would participate in organised sports. We measured motives using eight items on a 4-point Likert scale (1 = totally disagree; 4 = totally agree) that began with "Why do/would you participate in organised sports?". These items were developed based on an existing valid and reproducible Dutch questionnaire for adults [19] and previous studies on physical activity behaviours among adolescents [20, 21]. Due to limitations in the organisational aspect and to keep the burden for participants acceptable, we chose to include only a small selection of eight items from the previous studies. We made this selection of eight items based the Motives for Physical Activity Measure-Revised (MPAM-R) scale. This scale assesses the strength of five motives: enjoyment, social, competence/challenge, appearance, and fitness. It was introduced and validated by Ryan et al. (1997) [22] and uses the theoretical background of the Self-Determination Theory [23]. For each of the five motives, we included at least one item in our questionnaire. Some items were modified accordingly to fit the current study or to make them age appropriate. We included one item for the enjoyment ("it is a fun thing to do"), appearance ("to improve body shape"), and fitness ("to be healthy") motives, two items for the social motive ("to meet up with peers" and "friends participate too") and three items for the competence/challenge motive ("to perform well", "to be the best", "to compete with others"). Motives was calculated as the average of a student's responses on all items, with a Cronbach alpha of 0.75 indicating an acceptable level of reliability.

**Barriers.**   Barriers referred to the reasons why students do not or would not participate in organised sports. Students responded to 11 items on a 4-point Likert scale (1 = totally disagree; 4 = totally agree) that began with "Why do/would you not participate in organised sports?". Again, we developed the items based on the Dutch questionnaire for adults [19] and previous studies among children and adolescents [20, 21, 24, 25]. For measuring barriers, we included a selection of 11 items, some of which were adapted to fit the current study and the age of the study population. We included the following items: "I dislike it", "I am not good in doing sports", "it takes too much time", "it is too expensive", "I do not like the atmosphere", "I do not like training sessions", "I do not like competition", "I often get injured", "I get bullied", "my friends do not participate", and "I am not allowed by my parents". Barriers was calculated as the average of a student's responses on all items, with a Cronbach alpha of 0.70 indicating an acceptable level of reliability.

**Perceived competence.**   Perceived competence is defined as a student's perception of their competence in sports. Students responded to a single item "How good do you think you are at sports?" on a 4-point Likert scale (1 = not good at all; 4 = very good). While this single-item measure was used in this study to reduce participant burden caused by the questionnaire's length, it lacks formal validation from previous studies. Recognizing the limitations of a single-item approach, our longitudinal study adopted a more comprehensive measure of perceived competence from Wave 2 onwards. This measure was developed by adapting items from Harter's scale on physical competence [26]. Unfortunately, these refined measures were not available at Wave 1 and could therefore not be included in the current analyses. Despite its preliminary nature, the single-item measure demonstrated a significant correlation of .607 (p < .001) with the refined measurement.

**Encouragement.**   We conceptualised encouragement as the number of individuals who encourage students to participate in sports. Typically, encouragement is measured using items that ask participants about the frequency with which parents and/or friends encourage them to participate in physical activities or sports [20, 25]. In our study, we aimed to explore not only who encouraged students but also the diversity of these sources beyond parents and friends. Therefore, we adapted existing items from prior research to include additional significant sources of encouragement such as siblings, trainers/coaches, and physical education teachers, all recognized as crucial sources of social support in previous studies [27, 28]. It is important to note that while we adapted the encouragement items to encompass a broader range of sources, we did not conduct specific validation procedures for this modified tool in our study.

During Wave 1, students were asked whether anyone encouraged them to participate in organised sports. If affirmative, they specified which of the six persons encouraged. During Wave 2, students were asked who encouraged them from a list including the same six persons. They could choose more than one person from the list. The final score for encouragement was calculated based on the number of individuals selected by each student.

**Motor skills.**   In our study, motor skills referred to fundamental movement skills. Fundamental movement skills are basic (mainly gross motor coordinative) skills that form a basis for the more specific sports skills. We objectively assessed fundamental movement skills using a combination four test items: three test items of the KTK short form [29] (i.e. walking backwards, moving sideways and jumping sideways) and the Faber's eye hand coordination test [30]. This combination of test items was first proposed by Platvoet et al. (2018) [31] (a detailed protocol of all four tests can also be found in this study) and was validated by Coppens et al. (2021) [32]. All motor skill competence test items were assessed shoeless and in light clothing by a team of trained testers (including researchers and students from the Physical Education Teacher and Pedagogical Sciences education programs in Nijmegen, Netherlands). A composite score was computed by conducting a factor analysis with varimax rotation on the raw motor skills scores using SPSS software (version 27) [33]. This analysis included all motor skill items, and factor loadings were interpreted to calculate a factor score. The resulting factor score represents the composite score that captures the overall motor skills performance.

## Procedures

The team of trained testers collected data during a regular physical education class in an indoor facility. During the visit, students completed an online questionnaire on their laptop computers or smartphone via LimeSurvey. The questionnaire assessed participation in organised sports, motives, barriers, perceived competence, encouragement, age and ethnicity. The questionnaire is available at the following link: https://doi.org/10.17026/SS/LA3KIZ. In addition, during the same visit, the four motor skill tests were administered and anthropometric measurements were taken.

## Data analysis

All analyses were conducted using Stata software (version 17) [34]. To examine changes in the YPAP factors (i.e. motor skills, encouragement, perceived competence, barriers, and motives) between Wave 1, Wave 2 and Wave 3, we performed dependent $t$ tests (using the 'ttest' command in Stata) with Bonferroni correction.

**Participation in organised sports.**   To examine changes in participation in organised sports over time (first research question), we performed two types of analyses. First, we used a Cochran's $Q$ test (using the 'cochranq' command in Stata) to assess differences in the

proportions of students that participated in organised sports between Wave 1, Wave 2 and Wave 3. Second, we used a dependent *t* test (using the 'ttest' command in Stata) to assess changes in hours of participation per week between Wave 2 and Wave 3.

**Relationships between factors and participation in organised sports one year later.** Our data is characterised by a four-level structure, requiring multilevel modelling [35]. The data consisted of repeated measures (level 1) that are nested within adolescents (level 2) who are clustered in classes (level 3) within schools (level 4). Although the data was nested within schools, the number of schools (n = 5) was too low to perform four-level analyses [35]. Although the number of waves was also too low [35], in order to model within-person changes over time, it is still appropriate (e.g. [36, 37]) and needed to use time as a separate level even when the number of measurements is low [38, 39]. Therefore, we accounted for a three-level structure with repeated waves of data collection and adolescents nested in classes (n = 42).

To examine whether the factors predicted adolescents' weekly hours of participation in organised sports one year later (second research question), we conducted two multilevel linear fixed effects regression models (using the 'xtreg' command in Stata). We predicted participation in organised sports of Wave 2 and Wave 3 by the YPAP factors of Wave 1 (on Wave 2) and Wave 2 (on Wave 3). In Model 1, we modelled all background measures (i.e. age, gender, ethnicity, and BMI) and baseline participation (yes or no) as predictors of participation one year later. In Model 2, we added all YPAP factors simultaneously to Model 1.

To examine the differences between the relationships of each YPAP factor and weekly hours of participation in organised sports over time (third research question), we added interaction terms between each factor and wave of data collection. In Model 3a-e, we added one interaction term at a time (i.e. motor skills*Wave, encouragement*Wave, perceived competence*Wave, barriers*Wave, and motives*Wave, respectively). In Model 4, we included all interaction terms simultaneously into the model.

## Results

Table 1 presents the descriptive statistics for the background variables, participation in organised sports and the YPAP factors.

### Participation in organised sports

The proportion of adolescents that participated in organised sports on Wave 1, Wave 2 and Wave 3 was 86.8% (n = 177), 87.8% (n = 179) and 87.3% (n = 178), respectively. Cochran's *Q* test determined that there was no statistically significant difference in the proportion of adolescents who participated in organised sports over time, $\chi^2(2) = 0.176$, $p = .916$. Adolescents participated in organised sports for an average of 3.9 hours per week (SD = 2.7) on Wave 2 and for an average of 4.0 hours per week (SD = 2.9) on Wave 3. A dependent *t* test indicated that there was no statistically significant change in the hours per week that adolescents participated in organised sports, $t(203) = -1.174$, $p = .242$.

### Relationships between factors and participation in organised sports one year later

Table 2 presents the results of the multilevel linear fixed effects regression models. In Model 1, all background variables and baseline participation significantly predicted adolescents' hours of participation in organised sports. Boys and native Dutch adolescents spent more time in organised sports than girls (B = 0.82, *p* < .01) and non-native Dutch adolescents (B = 1.14, *p* < .01), respectively. In addition, older adolescents (B = -0.73, *p* < .05) and adolescents with a higher BMI (B = -0.12, *p* < .05) spent less time in organised sports. Finally, adolescents who

**Table 1. Descriptive statistics.**

| | Scale | Wave 1 | | | Wave 2 | | | Wave 3 | | |
| --- | --- | --- | --- | --- | --- | --- | --- | --- | --- | --- |
| | | % | M | SD | % | M | SD | % | M | SD |
| Background variables | | | | | | | | | | |
| Age | | | 12.46 | 0.44 | | | | | | |
| Boy | | 51.47 | | | | | | | | |
| Native | | 88.73 | | | | | | | | |
| BMI | | | 18.18[a,c] | 2.44 | | 19.00[a,b] | 2.75 | | 19.66[b,c] | 2.80 |
| Organised sports | | | | | | | | | | |
| Yes | | 86.76 | | | 87.75 | | | 87.25 | | |
| Hours per week | | | | | | 3.85 | 2.69 | | 4.03 | 2.92 |
| Factors | | | | | | | | | | |
| Walking backwards | 0–72 | | 47.38[a] | 13.68 | | 50.99[a,b] | 12.61 | | 46.44[b] | 15.76 |
| Jumping sideways | | | 73.27[a] | 11.76 | | 77.38[a,b] | 12.61 | | 69.96[b] | 27.41 |
| Moving sideways | | | 50.08[a,c] | 8.08 | | 53.58[a] | 8.41 | | 53.63[c] | 8.45 |
| Eye hand coordination | | | 38.69[a,c] | 13.35 | | 44.29[a,b] | 12.41 | | 48.60[b,c] | 16.19 |
| Encouragement | 0–6 | | 3.26 | 2.17 | | 3.12[b] | 1.66 | | 3.45[b] | 1.60 |
| Perceived competence | 1–4 | | 3.24[a,c] | 0.58 | | 3.15[a] | 0.61 | | 3.13[c] | 0.65 |
| Barriers | 1–4 | | 1.48 | 0.38 | | 1.45 | 0.41 | | 1.46 | 0.41 |
| Motives | 1–4 | | 3.35[a,c] | 0.49 | | 3.06[a] | 0.52 | | 3.11[c] | 0.55 |

Means in the same row with the same superscripts are significantly different from each other at $p < .017$ (Bonferroni-adjusted p-value). Number of observations on Wave 3 differ from the number of observations on Wave 1 and Wave 2 (n = 204) for the following variables: n = 196 (BMI), n = 188 (walking backwards), n = 187 (jumping sideways), n = 188 (moving sideways), and n = 192 (eye hand coordination).

participated in organised sports in the year before participated more hours in organised sports per week one year later (B = 2.59, $p < .001$).

In Model 2, encouragement, perceived competence and motives significantly predicted adolescents' hours of participation in organised sports, whereas motor skills and barriers did not predict adolescent's hours of participation. Adolescents with more encouragement (B = 0.17, $p < .05$), higher perceived competence (B = 0.62, $p < .05$) and higher motives (B = 0.79, $p < .05$) in the year before spent more hours in organised sports one year later. In addition, the background variables age (B = -0.71, $p < .05$), gender (B = 0.69, $p < .01$) and native (B = 1.05, $p < .01$) as well baseline participation (B = 1.51, $p < .001$) remained significant predictors in this model, while BMI was no longer a significant predictor of participation (B = -0.07, $p = .180$).

### Changes in the relationships between factors and participation in organised sports over time

In Models 3a-e (see Table 2), encouragement ($p < .05$), perceived competence ($p < .05$) and motives ($p < .05$) remained significant predictors of participation in organised sports. Also, the interaction terms of these models indicate that there were no significant changes over time in any of the relationships between the factors and participation (motor skills: B = 0.26, $p = .324$; encouragement: B = -0.03, $p = .818$; perceived competence: B = 0.08, $p = .857$; barriers: B = 0.26, $p = .684$; motives: B = -0.18, $p = .723$). Furthermore, the effects of age ($p < .05$), gender ($p < .01$), native ($p < .01$) and baseline participation ($p < .001$) remained significant in these models.

In Model 4, only encouragement (B = 0.18, $p < .05$) remained a significant predictor of participation in organised sports. Perceived competence (B = 0.55, $p = .140$) and motives

**Table 2. Multilevel linear fixed effects regression models.**

|  | Model 1 | Model 2 | Model 3a | Model 3b | Model 3c | Model 3d | Model 3e | Model 4 |
|---|---|---|---|---|---|---|---|---|
| Age | -0.73* | -0.71* | -0.70* | -0.71* | -0.71* | -0.70* | -0.71* | -0.69* |
|  | (0.33) | (0.31) | (0.31) | (0.32) | (0.32) | (0.32) | (0.32) | (0.32) |
| Boy = ref. | 0.82** | 0.69** | 0.69** | 0.69** | 0.69** | 0.69** | 0.69** | 0.70** |
|  | (0.27) | (0.27) | (0.27) | (0.27) | (0.27) | (0.27) | (0.27) | (0.27) |
| Native = ref. | 1.14** | 1.05** | 1.05** | 1.06** | 1.05** | 1.04** | 1.05** | 1.05* |
|  | (0.42) | (0.40) | (0.40) | (0.40) | (0.40) | (0.40) | (0.40) | (0.41) |
| BMI | -0.12* | -0.07 | -0.07 | -0.07 | -0.07 | -0.07 | -0.07 | -0.07 |
|  | (0.05) | (0.05) | (0.05) | (0.05) | (0.05) | (0.05) | (0.05) | (0.05) |
| Baseline participation | 2.59*** | 1.51*** | 1.54*** | 1.52*** | 1.51*** | 1.51*** | 1.51*** | 1.55*** |
|  | (0.41) | (0.43) | (0.43) | (0.43) | (0.43) | (0.43) | (0.43) | (0.43) |
| Wave 3 | 0.78 | 1.11** | 1.08* | 1.21* | 0.87 | 0.72 | 1.69 | 0.77 |
|  | (0.42) | (0.42) | (0.42) | (0.61) | (1.38) | (1.04) | (1.70) | (2.81) |
| Motor skills |  | 0.30 | 0.17 | 0.30 | 0.30 | 0.30 | 0.30 | 0.14 |
|  |  | (0.16) | (0.21) | (0.16) | (0.16) | (0.16) | (0.16) | (0.22) |
| Encouragement |  | 0.17* | 0.17* | 0.18* | 0.17* | 0.17* | 0.17* | 0.18* |
|  |  | (0.07) | (0.07) | (0.09) | (0.07) | (0.07) | (0.07) | (0.09) |
| Perceived competence |  | 0.62* | 0.61* | 0.62* | 0.58 | 0.62* | 0.62* | 0.55 |
|  |  | (0.26) | (0.26) | (0.26) | (0.34) | (0.26) | (0.26) | (0.37) |
| Barriers |  | -0.06 | -0.05 | -0.05 | -0.06 | -0.20 | -0.04 | -0.26 |
|  |  | (0.39) | (0.39) | (0.39) | (0.39) | (0.52) | (0.39) | (0.58) |
| Motives |  | 0.79* | 0.80* | 0.80* | 0.80* | 0.78* | 0.90* | 0.90 |
|  |  | (0.31) | (0.31) | (0.31) | (0.31) | (0.31) | (0.43) | (0.49) |
| Wave 3*Motor skills |  |  | 0.26 |  |  |  |  | 0.31 |
|  |  |  | (0.26) |  |  |  |  | (0.28) |
| Wave 3*Encouragement |  |  |  | -0.03 |  |  |  | -0.02 |
|  |  |  |  | (0.14) |  |  |  | (0.14) |
| Wave 3*Perceived competence |  |  |  |  | 0.08 |  |  | 0.11 |
|  |  |  |  |  | (0.42) |  |  | (0.50) |
| Wave 3*Barriers |  |  |  |  |  | 0.26 |  | 0.42 |
|  |  |  |  |  |  | (0.64) |  | (0.75) |
| Wave 3*Motives |  |  |  |  |  |  | -0.18 | -0.19 |
|  |  |  |  |  |  |  | (0.51) | (0.63) |
| Intercept | 11.46** | 6.23 | 6.12 | 6.16 | 6.38 | 6.38 | 5.88 | 6.17 |
|  | (4.31) | (4.28) | (4.28) | (4.29) | (4.36) | (4.30) | (4.39) | (4.56) |
| R-squared for between model | 0.10 | 0.23 | 0.22 | 0.22 | 0.23 | 0.22 | 0.23 | 0.21 |
| R-squared for within model | 0.18 | 0.27 | 0.27 | 0.27 | 0.27 | 0.27 | 0.27 | 0.28 |

Table presents the regression coefficients and standard errors in brackets. Significance levels

*$p < .05$

**$p < .01$

***$p < .001$.

(B = 0.90, $p$ = .064) no longer significantly predicted participation. Additionally, the interaction terms of this model were still not significant (motor skills: B = 0.31, $p$ = .279; encouragement: B = -0.02, $p$ = .899; perceived competence: B = 0.11, $p$ = .820; barriers: B = 0.42, $p$ = .578; motives: B = -0.19, $p$ = .759), indicating that there were no changes over time in any of the relationships between the factors and participation. Finally, the background variables age (B =

-0.69, $p < .05$), gender (B = 0.70, $p < .01$), native (B = 1.05, $p < .01$) and baseline participation (B = 1.55, $p < .001$) remained significant predictors in this final model.

## Discussion

This paper adds to the existing literature on participation in youth sports by prospectively examining adolescents' participation in organised sports and the potential factors that affect participation. First, we examined change in participation over two years from the start of secondary education. Second, we examined whether a unique combination of factors (i.e. motives, barriers, perceived competence, encouragement and motor skills) based on the Youth Physical Activity Promotion (YPAP) model [10] was associated with participation after one year. Although each of these factors has been studied into detail in cross-sectional and retrospective studies (see [7–9]), this combination of factors had never been studied prospectively. Third, as a further extension of existing studies, we examined whether the associations between each of the factors and participation in organised sports changed over time. Below we elaborate on the results in detail.

### Participation in organised sports

Our findings showed that participation in organised sports was stable during the first years of secondary education in a sample of Dutch adolescents. This finding appears contradictory to participation trends found in this age group in the Netherlands [5] and other Western countries (e.g. [4, 6]). In addition, the participation rates in our sample were rather high (87–88%) compared to national data (63%) [5]. These differences might partly be explained by the characteristics of the schools that agreed to participate in our study. Three of the five schools pay special attention to sports activities in their curriculum. This focus on sports presents a likely bias in our sample as it especially attracts adolescents who already participate in sports and/or those who are motivated towards sports. As indicated by previous studies [3, 40], participation in organised sports during childhood and adolescence can be represented by different trajectories. Some of the trajectory groups in these studies represented high and stable patterns of participation in organised sports (e.g. high stable and consistent participators), whereas other trajectory groups mainly represented lower and/or decreasing patterns (e.g. high decreasing, low decreasing, nonparticipators, and dropouts). Thus, although participation in organised sports clearly declines in many adolescents [4–6], our results reinforce that this does not apply to a specific part of the population. Moreover, and more speculatively, the school environment with its emphasis on sports may have significantly contributed to the absence of a decline. Finally, the high percentage of participation may introduce a ceiling effect with little room for further increase.

### Relationships between factors and participation in organised sports one year later

Our findings showed that a combination of individual and environmental factors predicted participation in organised sports after one year. Specifically, adolescents who reported higher motives, higher perceptions of competence and more encouragement spent more time in organised sports one year later.

Building on the insights of our previous work [15], the current study underscores the important role of motivation for adolescents' participation in organised sports [7–9, 11]. It appeared that the adolescents in the present study generally experienced a high degree of motives, and–in line with a recent study among Dutch adolescents [41]–a low degree of barriers for participation. However, we found no evidence to support the hypothesis that barriers

negatively predicted participation after one year. These findings suggest that having few barriers to participation does not positively influence participation, whereas having many reasons for participation does. It is therefore important that professionals in youth sports pay particular attention to stimulating motivation. Future studies are warranted to unravel the sources of motivation (i.e. enjoyment, social, competence, appearance, and fitness) and their relative strength into more depth [22].

In line with the literature [7–9], our findings also indicated that perceived competence was an important predictor of participation in youth sports. In contrast, however, actual competence (i.e. motor skills) did not predict participation. This appears contradictory to the literature (e.g. [9]). However, the findings of a previous study might offer an explanation for the importance of perceived competence over actual competence [42]. This study observed that adolescents who overestimated their competence displayed better motivation and were more involved in sports compared to their peers with similar levels of actual competence and who accurately perceived their competence. Moreover, adolescents with low actual competence who overestimated themselves even participated in equal amounts of organised sports when compared to adolescents with average levels of competence who either accurately estimated or overestimated themselves. This suggests that perceptions of competence–especially when they are positive–may have a greater impact on participation in organised sports than actual competence. In the present study, over ninety percent of the adolescents perceived themselves as being good (or very good) at sports while their performance on the motor skill tests suggested that part of them had average or low actual competence [32]. It is therefore reasonable to assume that a significant portion of the adolescents in our study overestimated their competence. Together, these findings emphasise that developing perceived competence is a crucial factor for participation in youth sports.

Our findings furthermore showed that adolescents who received encouragement from various people spent more time in organised sports one year later. It had already been shown that–in certain sports–the amount of parental, peer and coach support predicts persistent participation over time [9]. Also, in cross-sectional studies, the combined effects of support by different people have been shown to play a significant role in youth physical activities (see [27, 43]). Thus, extending these studies, our findings suggest that being supported by different people in different contexts (e.g. home, sports, school) is important for adolescents' participation in organised sports over time. It is still possible that, over time, the same persons encourage an adolescent to participate in sports, but that the extent to which these persons encourage the adolescent does change over time (e.g. parents provide less support and friends provide more support) [44]. Similarly, different adolescents may receive encouragement from the same (number of) persons, but the extent to which they receive encouragement can differ between adolescents [43]. Further research focused on unravelling the different sources of encouragement (e.g. parents, peers, coach) is therefore much needed. In addition, there may be promise in examining other types of social support (e.g. instrumental support, co-participation) in relation to participation in organised sports [27, 28].

## Changes in the relationships between factors and participation in organised sports over time

This study was the first to examine changes in the relationships between factors and participation in organised sports over time. Interestingly, the relationships of each of the factors (i.e. motives, barriers, perceived competence, encouragement, and motor skills) with participation remained stable over time. This is interesting since adolescence is considered a key developmental period that involves significant changes related to the individual (e.g. start of puberty)

[45] as well as the environment (e.g. new school, new peer groups) [46]. These changes likely also have an impact on participation in sports and its related factors [45]. As such, we did observe changes in some of the factors. For example, the scores on both perceived competence and motives slightly declined while the scores on the four motor skills tests fluctuated. Nevertheless, our findings suggest that this not necessarily implies that the role of these factors in participation in organised sports changes. This would highlight the importance to enhance motives, perceived competence and encouragement in youth sport contexts already at a young age. It should, however, still be acknowledged that this study was the first to examine changes in the relationships between factors and sports over time and that our sample may have been biased towards individuals who are motivated to engage in sports. Therefore, these findings must be interpreted with caution.

## Limitations

Limitations of the current study include the high percentage of adolescents who continuously participated in organised sports [5] and the low percentage of adolescents who followed pre-vocational secondary education [47]. These aspects limit the generalisability of the findings. The aspect of the educational tracks is particularly important because, in the Netherlands, students in pre-vocational secondary education are less likely to participate in sports clubs than their peers in the other educational tracks [48]. This may partly explain the high rates of participation in organised sports in the present study. It is therefore warranted to replicate this study with a more representative sample (i.e. to attract more adolescents that do not participate in sports and/or are enrolled in preparatory secondary vocational education). One sampling strategy that should at least be included is to organise information meetings for all parents and children at the participating schools. Unfortunately, we were unable to organise such information meetings because of the COVID-19 restrictions at the start of the study.

Another study limitation includes the possible low sensitivity of some of our measures to detect differences between individuals and observations. For example, a very high percentage of the participants reported a Likert scale score of 3 or 4 on the perceived competence-item. Although this may have influenced our results, we believe that our decisions to make the length of the questionnaire acceptable were necessary to keep participants' motivation for the study as high as possible during a wave of data collection as well as over all waves of data collection. Future studies could, however, consider to spread the assessment of a questionnaire over two days. In that way, each of the measures can be assessed more thoroughly.

## Future research

To our knowledge, the present study was the first to prospectively examine the effects of various factors of the YPAP model. Since we only included the direct effects of the model, future research would benefit from also prospectively examining the indirect effects of YPAP factors at multiple levels of influence. The relevance of these indirect effects has been pointed out in several cross-sectional studies [49–51]. Furthermore, corroborating previous research (e.g. [7, 9, 11, 15]), the current study confirms the important roles of motives, perceived competence and encouragement for participation in organised sports. Additional next steps for future research would be to unravel the extent to which each motive (e.g. enjoyment, competence, social, appearance, and fitness) and the extent to which each of the different sources of encouragement (e.g. family, friends, coaches) contribute to participation in and dropout from organised sports. Finally, next to modifiable factors such as motives, perceived competence, and encouragement, future research should always take into account demographic and non-modifiable factors such as ethnicity and gender. The findings of the present study indicated that

these non-modifiable factors played a significant role in predicting participation in organised sports. Furthermore, since our study could not explain roughly 80% of the variance in participation, it is crucial to explore additional modifiable factors that may influence this outcome and to account for individual variability. This may inform interventions as it helps to identify the individuals that would benefit most from an intervention in addition to the factors that should be targeted with the intervention.

## Conclusion

This study contributes to our understanding of youth participation in organised sports. Although participation in organised sports regularly declines during adolescence, the findings of the current study suggest that participation in organised sports can be stable during the first years of secondary education for a subgroup of adolescents. Moreover, we showed that students who reported more encouragement, higher perceptions of competence and higher levels of motives spent more time in organised sports after one year. Therefore, coaches, teachers and parents in youth sport contexts should focus on stimulating levels of motivation, perceived competence as well as encouragement. This should be done already during primary education, since the findings indicated that the relationships between such factors and participation remained stable during the first years of secondary education. Future research should unravel the sources of, especially, motivation and encouragement as well as the indirect relationships of the factors with participation. This would further inform interventions strategies that stimulate continuous participation in and prevent dropout from youth sports.

## Acknowledgments

First, we would like to thank Jolien Maas of the Behavioural Science Institute, Radboud University, Nijmegen, Netherlands for their valuable assistance in organising the data collection of this study. We would also like to thank Sebastiaan Platvoet and Mark de Niet of the Institute for Studies in Sports and Exercise, HAN University of Applied Sciences, Nijmegen for their contributions to various aspects of this research project. Finally, we would like to acknowledge the students of the Radboud University and HAN University of Applied Sciences who collected the data, and the schools, PE teachers, and students who participated in the study.

## Author Contributions

**Conceptualization:** Gwennyth E. Spruijtenburg, Femke van Abswoude, Hidde Bekhuis, Bert Steenbergen.

**Data curation:** Gwennyth E. Spruijtenburg.

**Formal analysis:** Gwennyth E. Spruijtenburg, Hidde Bekhuis.

**Funding acquisition:** Femke van Abswoude, Hidde Bekhuis, Bert Steenbergen.

**Investigation:** Gwennyth E. Spruijtenburg.

**Methodology:** Gwennyth E. Spruijtenburg, Femke van Abswoude, Bert Steenbergen.

**Project administration:** Gwennyth E. Spruijtenburg.

**Supervision:** Femke van Abswoude, Bert Steenbergen.

**Validation:** Gwennyth E. Spruijtenburg, Femke van Abswoude, Hidde Bekhuis, Bert Steenbergen.

**Visualization:** Gwennyth E. Spruijtenburg.

**Writing – original draft:** Gwennyth E. Spruijtenburg.

**Writing – review & editing:** Gwennyth E. Spruijtenburg, Femke van Abswoude, Hidde Bekhuis, Bert Steenbergen.

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
