## [Decision Letter · Decision Letter 0]

11 Jul 2024

PONE-D-24-19598Factors predicting participation in organised sports during adolescence: A two-year longitudinal perspectivePLOS ONE

Dear Dr. Spruijtenburg,

Thank you for submitting your manuscript to PLOS ONE. After careful consideration, we feel that it has merit but does not fully meet PLOS ONE’s publication criteria as it currently stands. Therefore, we invite you to submit a revised version of the manuscript that addresses the points raised during the review process.

We look forward to receiving your revised manuscript.

Kind regards,

Henri Tilga, PhD

Academic Editor

PLOS ONE

Journal Requirements:

"This research was supported by a grant (TRIAL) from the Dutch Research Council (NWO), grant number NWA.1160.18.249. "

4. Thank you for uploading your study's underlying data set. Unfortunately, the repository you have noted in your Data Availability statement does not qualify as an acceptable data repository according to PLOS's standards.

Additional Editor Comments:

The Reviewers have provided detailed suggestions to further improve the quality of this manuscript.

Reviewers' comments:

Reviewer's Responses to Questions

**Comments to the Author**

1. Is the manuscript technically sound, and do the data support the conclusions?

Reviewer #1: Yes

2. Has the statistical analysis been performed appropriately and rigorously? 

Reviewer #1: Yes

3. Have the authors made all data underlying the findings in their manuscript fully available?

Reviewer #1: Yes

4. Is the manuscript presented in an intelligible fashion and written in standard English?

Reviewer #1: Yes

5. Review Comments to the Author

Reviewer #1: Thank you for the opportunity to review the manuscript entitled “Factors predicting participation in organised sports during adolescence: A two-year longitudinal perspective”.

This paper examined change in students’ participation over two years from the start of secondary Education and whether a unique combination of factors (i.e. motives, barriers, perceived competence, encouragement and motor skills) based on the Youth Physical Activity Promotion (YPAP) model was associated with participation after one year. This paper presents original data and is well-written. The authors clearly presented the introduction, method and results. I think the paper is interesting and in line with the aims of the journal, although I have some comments I hope help the authors to improve the quality of their document before its publication.

Comment 1: Lines 115-116. Is there a protocol paper published? If yes, I recommend to include the information.

Comment 2: Line 119. Even when researchers have done 7 waves of data collection, I humble consider that if the paper is focused on 3 waves, only the information about that 3 waves should appear. Please, consider to change “seven waves” for “three waves” and change the number of the waves (should be 1, 2, and 3, instead of 1, 4 and 7).

Comment 3: Lines 138. Please, include further information about drop out rates.

Comment 4: Lines 199-201. Is the perceived competence single item validated or used in previous studies? Please include such information. I am wondering how one item could offer valid information about perceived competence.

Comment 5: Lines 202-214. Authors mentioned a modification included in the encouragement tool/questionnaire used. How the authors validated that tool? How the information obtained is used in the posterior analysis? They mentioned that multiple response were allowed so the response was categorized and used the number of persons that encourage students? Please, clarify this point.

Comment 6: Please, indicate who assess the motor competence test. I assume that was a researcher, but this information should appear explicitly. Also indicate if participants performed 1 or 2 trials per test and how the motor quotient was calculated.

Comment 7: I am not an expert in the analysis conducted however I would like to know if an R2 of .20 is considered a good value considering the number of variables included in the models (table 2).

6. PLOS authors have the option to publish the peer review history of their article (what does this mean?). If published, this will include your full peer review and any attached files.

Reviewer #1: No

---

## [Author Response · Author response to Decision Letter 0]

6 Aug 2024

Reviewers' comments

Reviewer #1: Thank you for the opportunity to review the manuscript entitled “Factors predicting participation in organised sports during adolescence: A two-year longitudinal perspective”.

This paper examined change in students’ participation over two years from the start of secondary Education and whether a unique combination of factors (i.e. motives, barriers, perceived competence, encouragement and motor skills) based on the Youth Physical Activity Promotion (YPAP) model was associated with participation after one year. This paper presents original data and is well-written. The authors clearly presented the introduction, method and results. I think the paper is interesting and in line with the aims of the journal, although I have some comments I hope help the authors to improve the quality of their document before its publication.

Response: Thank you for your thoughtful and positive feedback on our manuscript. We appreciate your acknowledgment of the originality and clarity of our work, as well as your recognition of its alignment with the aims of the journal. We have carefully considered your comments and have made the necessary revisions to enhance the quality of our manuscript. We believe these changes have strengthened our paper, and we are grateful for your constructive suggestions. Thank you once again for your time and valuable input.

Comment 1: Lines 115-116. Is there a protocol paper published? If yes, I recommend to include the information.

Response: Although the data collection protocol is thoroughly described in the main manuscript, there is no protocol paper published for this study. 

Comment 2: Line 119. Even when researchers have done 7 waves of data collection, I humble consider that if the paper is focused on 3 waves, only the information about that 3 waves should appear. Please, consider to change “seven waves” for “three waves” and change the number of the waves (should be 1, 2, and 3, instead of 1, 4 and 7).

Response: We appreciate your suggestion to focus specifically on the three waves of data collection used in this study. We revised the manuscript accordingly, replacing references to "seven waves" with "three waves" and adjusting the wave numbers to clearly indicate Waves 1, 2, and 3 where applicable.

Comment 3: Lines 138. Please, include further information about drop out rates.

Response: Thank you for highlighting the need for further information about dropout rates. We have now included detailed information about dropout rates in the revised manuscript (Lines 139-144). Initially, we had complete data for 390 students at Wave 1. By Wave 2 (in the previous version Wave 4), this number decreased to 285 students (73.1% of the original sample). By Wave 3 (in the previous version Wave 7), the number of students with complete data further decreased to 204 (52.3% of the initial sample). The dropout rates were 26.9% by Wave 2 and 47.7% by Wave 3. Common reasons for dropout included repeating a grade, moving to a different location, no longer wanting to participate, and being ill on the day of testing. We have included this information to provide a clearer understanding of participant retention throughout the study.

Comment 4: Lines 199-201. Is the perceived competence single item validated or used in previous studies? Please include such information. I am wondering how one item could offer valid information about perceived competence.

Response: The single item used to measure perceived competence in our study has not been validated nor used in previous studies. We acknowledge the concern about the validity of using a single item to assess perceived competence. Our primary rationale for adopting this approach was to minimize participant burden, given the extensive length of our questionnaire. However, we recognize that this justification has limitations.

Because of the limitations of a single-item measure, we improved our approach to measuring perceived competence from Wave 2 onwards. Unfortunately, data from these improved measures were not available at Wave 1, so we could not include them in the current analyses.

From Wave 2 onwards, the improved measurement of perceived competence was based on adapting and translating items from Harter's scale on physical competence. These items were adjusted to focus specifically on sports-related perceptions of physical competence among adolescents.

Despite the lack of formal validation, it is notable that the single-item measure shows a significant correlation of .607 (p < .001) with the improved measurement of perceived competence used in the later phases of the study. This suggests that, while the single-item measure has its limitations, it provides a useful but basic assessment of perceived competence among participants.

In the revised manuscript, we have included the correlation analysis to offer additional context on the validity of our measurement approach (Lines 210-217). We acknowledge the importance of future validation efforts to enhance the reliability of perceived competence assessments in similar studies.

Comment 5: Lines 202-214. Authors mentioned a modification included in the encouragement tool/questionnaire used. How the authors validated that tool? How the information obtained is used in the posterior analysis? They mentioned that multiple response were allowed so the response was categorized and used the number of persons that encourage students? Please, clarify this point.

Response: Thank you for your comments. We acknowledge your concerns regarding the validation and use of this tool. Regarding validation, we modified the encouragement items from prior research to encompass a broader spectrum of influential individuals beyond parents and friends, such as siblings, trainers/coaches, and physical education teachers, based on their recognized role in social support. While these modifications were informed by literature, specific validation procedures for this adapted tool were not conducted in our study. This is an important consideration, and we have added this clarification in the revised manuscript (Lines 219-232).

In terms of how this information was used in our analysis, during Wave 1, students were asked about encouragement from various sources using structured items. If affirmative, they specified which individuals encouraged them. At Wave 2 students were asked to indicate who encouraged their sports participation from a list including the same six categories. Multiple responses were allowed, and the final encouragement score was computed based on the number of individuals selected by each student. We have included additional details in the revised manuscript (Lines 233-241) to explain these procedures.

Comment 6: Please, indicate who assess the motor competence test. I assume that was a researcher, but this information should appear explicitly. Also indicate if participants performed 1 or 2 trials per test and how the motor quotient was calculated.

Response: We have revised the manuscript to explicitly state that the motor competence tests were assessed by a team of trained testers, including researchers and students from the Physical Education Teacher and Pedagogical Sciences education programs at Radboud University in Nijmegen, Netherlands (Lines 250-252).

We have chosen not to specify in the manuscript whether participants performed 1 or 2 trials per test, as the detailed protocols for each test are extensively documented in another related study (which is already referenced in the text in Line 248). However, we are prepared to include this information if you believe it would enhance the manuscript.

In response to your comment about how the motor quotient was calculated, we have elaborated in the manuscript that a factor analysis with varimax rotation on the raw motor skills scores was conducted using SPSS (Lines 252-256). This analysis encompassed all motor skill items, and factor loadings were interpreted to compute the composite factor score representing overall motor skills performance. We do not have a motor quotient because there are no normative values for this test and age group available in the Netherlands.

Comment 7: I am not an expert in the analysis conducted however I would like to know if an R2 of .20 is considered a good value considering the number of variables included in the models (table 2).

Response: Since we are using multilevel regression analyses, the presented (adjusted) R2 is not the absolute explained variance, but an estimate. Nevertheless, an R2 of 0.20 means that our model accounts for roughly 20% of the variability in hours of participation in organized sports. Although this seems to be quite low, in social and behavioral sciences, such a value is pretty high and more than acceptable due to the complexity and multitude of influencing factors. Considering the 10 predictors in our model, including background variables (age, gender, ethnicity, BMI, baseline participation) and independent variables (motor skills, encouragement, perceived competence, barriers, motives), we believe an R2 of 0.20 is meaningful and reflective of the relationships captured in our study. Nevertheless, it would be valuable to aim for a higher R2 to explain more of the variability in participation in organised sports. Therefore, we have added a discussion in the revised manuscript (Lines 486-488) emphasizing the need to explore additional modifiable factors and account for individual variability in future research. This approach could help uncover more variables that influence participation and potentially improve the model’s explanatory power.

Editor Comments

The Reviewers have provided detailed suggestions to further improve the quality of this manuscript.

Response: Thank you for your message and for coordinating the review process for our manuscript. We appreciate the detailed suggestions provided by the Reviewers. Their insights have been invaluable, and we have thoroughly addressed each of their comments to further improve the quality of our manuscript. Additionally, during the revision process, we discovered an error in Table 2 related to the values of the 'R-squared for within model' and have corrected it in the manuscript. We want to assure you that this correction did not alter our interpretations or the overall conclusions of the study. Thank you once again for your guidance and support throughout this process.

---

## [Decision Letter · Decision Letter 1]

8 Sep 2024

Factors predicting participation in organised sports during adolescence: A two-year longitudinal perspective

PONE-D-24-19598R1

Dear Dr. Spruijtenburg,

We’re pleased to inform you that your manuscript has been judged scientifically suitable for publication and will be formally accepted for publication once it meets all outstanding technical requirements.

Kind regards,

Henri Tilga, PhD

Academic Editor

PLOS ONE

Additional Editor Comments (optional):

Reviewers' comments:

Reviewer's Responses to Questions

**Comments to the Author**

1. If the authors have adequately addressed your comments raised in a previous round of review and you feel that this manuscript is now acceptable for publication, you may indicate that here to bypass the “Comments to the Author” section, enter your conflict of interest statement in the “Confidential to Editor” section, and submit your "Accept" recommendation.

Reviewer #2: All comments have been addressed

Reviewer #3: (No Response)

2. Is the manuscript technically sound, and do the data support the conclusions?

Reviewer #2: Yes

Reviewer #3: Yes

3. Has the statistical analysis been performed appropriately and rigorously? 

Reviewer #2: Yes

Reviewer #3: Yes

4. Have the authors made all data underlying the findings in their manuscript fully available?

Reviewer #2: Yes

Reviewer #3: Yes

5. Is the manuscript presented in an intelligible fashion and written in standard English?

Reviewer #2: Yes

Reviewer #3: Yes

6. Review Comments to the Author

Reviewer #2: Authors have done well job on revising their manuscript. I think this manuscript is ready for the publication.

Reviewer #3: PONE- D-24-19598R1

Thank you for the opportunity to read this manuscript. The authors attempted to learn about the factors of participation in organized sports in a group of high school students. Although the research was supposed to combine combinations of psychosocial and motor factors, the research was mainly focused on psychosocial changes.

From the formal side: the layout of the work, formulation of the aim of the work, methods and research material were prepared correctly. The results were presented in a comprehensible and readable way.

The authors drew attention to the fact of high rates of participation in sports (in their research) in comparison to the national population, at the same time seeking answers to the observed phenomenon. In my opinion, this indicates the authors' great inquisitiveness and willingness to seek answers to questions that emerged during the research.

The authors showed that the combination of individual and environmental factors determined further participation in sports activity after a year. Among these factors, the main predictor of undertaken physical activity was motivation. Therefore, it seems understandable to focus the analysis of motivation on a broader group of influence, i.e.: family, peer group, coach, etc.

To sum up, the manuscript brings new elements of knowledge about factors influencing youth participation in organized physical activity, I propose to accept the prepared manuscript in its current form.

PhD Artur Kruszewski

7. PLOS authors have the option to publish the peer review history of their article (what does this mean?). If published, this will include your full peer review and any attached files.

Reviewer #2: No

Reviewer #3: **Yes: **Artur Kruszewski

---

## [Editor Report · Acceptance letter]

11 Sep 2024

PONE-D-24-19598R1 

PLOS ONE

Dear Dr. Spruijtenburg, 

I'm pleased to inform you that your manuscript has been deemed suitable for publication in PLOS ONE. Congratulations! Your manuscript is now being handed over to our production team.

Kind regards, 

on behalf of

Dr. Henri Tilga 

Academic Editor

PLOS ONE